# Companion restrictions in the emergency department during COVID-19: physician perceptions from the Western Cape, South Africa

Lauren E Wiebe ,[1] Helle Molsted Alvesson,[1] Willem Stassen [2]

¹Department of Global Public Health, Karolinska Institutet, Stockholm, Sweden
²Division of Emergency Medicine, University of Cape Town, Cape Town, South Africa

**Correspondence to**
Dr Helle Molsted Alvesson;
helle.molsted-alvesson@ki.se

## ABSTRACT

**Objectives** To determine emergency department (ED) physicians' perceptions regarding hospital companions being prohibited from accompanying the patient during COVID-19.

**Design** Two qualitative datasets were combined. Data collected included voice recordings, narrative interviewing and semistructured interviews. A reflexive thematic analysis was conducted and guided by the Normalisation Process Theory.

**Setting** Six hospital EDs in the Western Cape, South Africa.

**Participants** Convenience sampling was used to recruit a total of eight physicians working full time in the ED during COVID-19.

**Results** The lack of physical companions provided an opportunity for physicians to assess and reflect on a companion's role in efficient patient care. Physicians perceived that the COVID-19 restrictions illuminated that patient companions engaged in the ED as providers contributing to patient care by providing collateral information and patient support, while simultaneously engaging as consumers detracting physicians from their priorities and patient care. These restrictions prompted the physicians to consider how they understand their patients largely through the companions. When companions became virtual, the physicians were forced to shift how they perceive their patient, which included increased empathy.

**Conclusion** The reflections of providers can feed into discussions about values within the healthcare system and can help explore the balance between medical and social safety, especially with companion restrictions still being practised in some hospitals. These perceptions illuminate various tradeoffs physicians had to consider throughout the pandemic and may be used to improve companion policies when planning for the continuation of the COVID-19 pandemic and future disease outbreaks.

## INTRODUCTION

Emergency conditions represent approximately 90% of the global mortality.[1] High functioning emergency departments (EDs)

### STRENGTHS AND LIMITATIONS OF THIS STUDY

⇒ COVID-19 caused this study to pivot from purposive sampling to convenience sampling, potentially introducing a sampling bias affecting transferability and trustworthiness of the results.

⇒ One professional base was included, potentially introducing a physician bias, which can limit the transferability of this study.

⇒ In-depth knowledge of one professional base was sampled at different time points between August 2020 and January 2022, providing a sufficient period to capture changed practices.

⇒ By combining and comparing two datasets with different types of data, we were able to capture a wider range of perceptions, reinforcing our situational and reflexive data. This has strengthened the quality and dependability of our analysis and results in relation to our research aim.

are vital to alleviating this burden, but their operations are influenced by the greater environment, social context and disease burden.[1–3]

A variety of factors contribute to successful EDs. The WHO provides Essential Trauma Care Guidelines and emphasises the importance of the implementation of standard policies and protocols in acute care facilities.[4] Policies and protocols take time to establish, and the tumultuous nature of COVID-19 posed challenges for maintaining and adapting standardised procedures.[5] Additionally, EDs often face challenges with communication and mismanaged information. Health professionals in EDs spend 80% of their time engaging with communication processes, including with colleagues, management, patients and companions.[6] In a high-risk, fast-paced environment such as EDs, having established communication and information pathways is critical to patient care; given that up to 12% of errors in EDs are attributed to inadequate communication.[6–8]

Prior to COVID-19, patient companions were prominent participants of communication in EDs. The literature highlights the strengths and challenges associated with this.[9–13] Studies show companions interfering with patient care by causing interruptions for the medical team, potentially leading to clinical errors, as well as increasing the stress on healthcare professionals when performing life-saving procedures.[9 10] However, communication with companions is vital for gaining a quick medical history to inform effective care.[9 10] Furthermore, the companion is a key component of support for both the patient and family, resulting in greater patient motivation for recovery, reduced length of stay and less fear and anxiety for both the patient and family.[9 11–13] The literature published prior to COVID-19 showed mixed findings on health professionals' perspective of patient companions. Many studies found nurses more in favour of familial presence than physicians, especially in emergency situations such as resuscitation.[10 12]

During COVID-19, to reduce transmission rates, countries globally implemented policies that limited, if not completely restricted companions or escorts accompanying hospitalised patients, which resulted in both positive and negative outcomes.[14 15] Studies have shown various impacts on the patient such as mental health challenges as well as reduced hygiene and nutritional status.[14–16] Companions and escorts also experienced many emotional and mental consequences including depression, fear and anger.[15] Lastly, providers were impacted by the increasing demands for communication from families and the stress of facilitating virtual communication.[15 17] However, studies also reported positive impacts including less crowding, increased space for social distancing and reduced transmission among patients, companions and providers.[18 19] Overall, most existing studies do not support universal restrictive policies in clinical settings during COVID-19, on the basis that the risks outweigh the benefits. The primary risks identified include psychological deficits to the patient and family, worsened patient health outcomes, distress in connection with end-of-life care and increased stress for providers.[14–16 20] Additionally, the evidence of reduced transmission rates is limited in quality.[14–16 20]

As of June 2022, South Africa has had approximately 4 million cases and 102 000 COVID-19-related deaths.[21] The impacts of COVID-19 in South Africa, specifically the Western Cape, are vast, including increased mortality of COVID-19 and other diseases, delayed health-seeking behaviour and changes in injury presentation.[22] EDs are uniquely impacted by COVID-19 and companion restrictions because they frequently serve already vulnerable populations.[5] Additionally, emergent conditions are dynamic in nature, making it harder to consistently implement new and evolving COVID-19 regulations.[5]

Although there is existing research on the broad COVID-19 impacts, the effects of companion restrictions throughout the COVID-19 pandemic have been minimally explored, especially outside paediatric and palliative settings.[15 16 19] In addition, the perspective of emergency health providers is rarely included. The study aim was to determine ED physicians' perceptions regarding hospital companions and escorts being prohibited from accompanying the patient during COVID-19.

## METHODS

South Africa is known for having some of the highest numbers of trauma in the world, in addition to experiencing a quadruple burden of disease.[1 23 24] This study is specifically taking place in the Western Cape, South Africa, because a high proportion of the countries' emergent conditions and diseases present in EDs here.[23] It was also the first province within South Africa to experience a COVID-19 surge and has maintained a relatively high rate of cases and deaths, with approximately 21% of cases and 22% of COVID-19-related deaths in the country.[25] COVID-19 affected practices that contribute to successful care in South African EDs, such as companion restrictions, structurally splitting the ED into COVID-19 likely and COVID-19 unlikely, reduction of blood supply, shortage of personal protective equipment and the redistribution of health professionals leaving many departments with insufficient human resources.[26 27] The high number of emergent conditions and excessive burden of COVID-19 made it a unique and appropriate setting to evaluate the impacts of COVID-19.

### Study design

The study used two different qualitative datasets. Dataset I included data from the end of the first wave of COVID-19 in August to October 2020 and explored emergency medical personnel's lived experiences with the pandemic.[28] Dataset II was collected during the second and third waves, from February 2021 to January 2022, and included semistructured interviews of physicians' perceptions on how COVID-19 affected emergency care.

### Sampling and recruitment

Both datasets included in-hospital emergency personnel in the Western Cape. Convenience sampling was used, and emergency medicine joint staff were contacted using their registered email addresses in the University of Cape Town staff repository. Staff were asked to distribute the study invitation to their employees for equal access. Subsequent participants were collected through snowball sampling. Our proposed sample size was 12 participants, but our priority was reaching information saturation, which was assessed throughout the data collection process. Our final sample included eight participants, who worked full time in the ED throughout COVID-19, from six different hospitals.

The mean age of participants was 35 years, with 9 years of medical experience. Demographic information was gained through a verbal questionnaire. The sample characteristics are included in table 1 while maintaining anonymity (table 1).

**Table 1** Participant ID codes

| ID code | Self-identified gender | Job title | Years in profession (since qualification) |
|---|---|---|---|
| ID1 | Male | Medical officer | 9 |
| ID2 | Female | Consultant | 11 |
| ID3 | Female | Consultant | 5 |
| ID4 | Female | Medical officer | 11 |
| ID5 | Female | Registrar | 6 |
| ID6 | Male | Medical officer | 8 |
| ID7 | Male | Registrar | 9 |
| ID8 | Female | Registrar | 8 |

### Data collection

This study aimed to answer two questions: (1) Which practices in the ED do the physicians perceive as being changed by the restrictions on companions/escorts throughout COVID-19 and (2) how have the restrictions impacted how physicians conceptualise the role of companions in patient care? Dataset I included WhatsApp voice recordings and one-on-one online interviews, conducted in English or Afrikaans. Only physicians from this dataset were included. Voice recordings ranged from 17 s to 30 min, and interviews ranged from 40 to 60 min. In dataset II, a semistructured interview guide (online supplemental appendix 1) was used in all interviews and was inspired by the Normalisation Process Theory (NPT). The interview guide was piloted for clarity with an emergency physician in Sweden and was adjusted accordingly. The interviews were conducted online in English and ranged from 40 to 60 min.

### Patient and public involvement

We had no patient/public involvement.

### Analysis

The study aim and questions were inspired by Reflexive Monitoring, which has been identified as an important feature in qualitative improvements, as it highlights how people assess the impacts of new practices on themselves and others.[29] In our context, we are examining how physicians assess the impacts of companion restrictions in their personal practice, along with the appraisal of their colleagues, companions and the patient. Reflexive Monitoring is a part of the NPT, which focuses on the processes behind changes in thinking, practices, policies and organisation in healthcare settings, specifically resource-constrained contexts[29] (figure 1).

The data from both studies were subjected to an inductive, reflexive thematic analysis using the six steps as described by Braun and Clarke.[30] NVivo V.12 Pro (QSR International, Australia) was used to develop the main categories and themes. Transcripts were initially coded by one data collector and a list of latent codes was developed. After the interviews were coded, subcategories were discussed in relation to emerging themes and were narrowed down to identify the main theme(s). These steps were done as an iterative process, where the codes and emerging themes were continuously reviewed among the research team, to help achieve triangulation and comprehensive analysis.[31] Throughout the analysis, the data was reviewed in relation to reflexive monitoring in the NPT. However, this was used to reinforce, not determine, or limit the themes. Anonymity was prioritised in this study and each participant was given an ID code to ensure their information was not linked to their responses or quotations. Pseudonyms were used in the transcriptions.

### Reflexivity

The research group was made up of an interdisciplinary team, all with experience with qualitative studies in Cape Town. In addition, one of our primary investigators has extensive research experience within emergency medicine. In qualitative studies, the interviewer's position and relationship with participants can impact the comfort and openness of participants.[32] The interviewer was a master's student of public health, with no ties to the hospitals that the physicians worked at. This potentially contributed to building trust and the physicians feeling more comfortable to share information about their workplace.

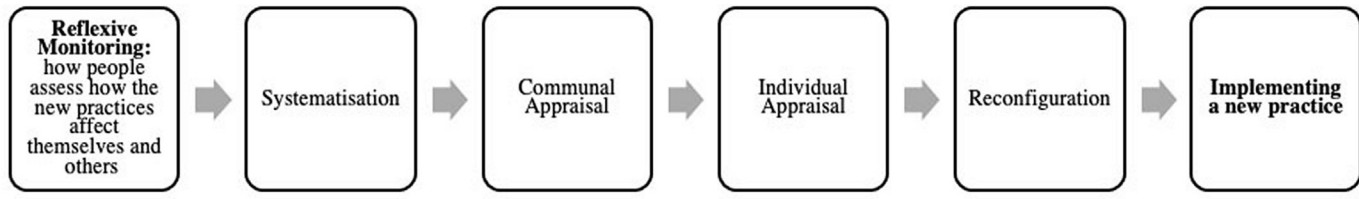

**Figure 1** Visual representation of the Normalisation Process Theory.

**Table 2** Results table

| Theme | Reconceptualising companions in the emergency department | | |
|---|---|---|---|
| Category | Virtual companions | | Finding balance |
| | Companions as providers | Companions as consumers | |
| Codes | Patient support Collateral information Nature of communication Provider empathy | Interruptions Crowding Call volume Time consumption | Visiting hours Technology Contact sheets |

## RESULTS

During COVID-19, the restriction was implemented that no companion was allowed to accompany the patient on arrival to the ED to reduce the risk of COVID-19 transmission.[27] In the absence of physical companions, practices surrounding patient support, physician and patient communication with companions and collection of collateral information were disrupted as companions became virtual. Based on physician perceptions, this change illuminated the different roles physical companions hold in patient care as both providers contributing to patient care, and consumers of care, potentially detracting attention from the patient in the ED (table 2).

### Companions as providers in the ED

Pre-COVID-19, physicians relied on patients being physically accompanied to the ED to be able to communicate with companions. Physicians acknowledged that companions play an important role for both the patient and the provider in the ED. When companions were not physically present, it often resulted in the patient being dependent on the health provider to communicate with the family on their behalf.

> A lot of them [patients] would not necessarily have a cell phone or a charged cell phone, or data to contact the families. And so, there were a lot of them that were very anxious and that definitely made it difficult for us.—ID1

The physicians expressed that the prominent negative aspect of not having physical companions were the challenges associated with obtaining collateral patient information relevant for clinical care, including pre-existing conditions, family history and medications. They viewed this as problematic to their ability to provide care.

> Having a physical escort there is often the only way to get collateral about a patient, and especially during COVID, so much about your decision about the level of care, and level of intervention was based on a patient's baseline … when these people didn't answer their phones, or you didn't have those numbers, it was literally impossible to make these kinds of decisions.—ID5

Physicians expressed that getting a hold of the companions was often time-consuming and occasionally, they could not get a hold of them at all, meaning they were having to make decisions regarding care without that critical information. Physicians also reflected on companion presence pre-COVID-19 and acknowledged that the physical presence resulted in higher quality and more individualised care planning, compared with trying to get a hold of companions virtually.

> This meant a lot more digging in folders looking for ID numbers, trying to contact family members outside of the hospital, phones that weren't answered, that really made an impact on how we manage patients and it also delayed turnaround time. Which meant we had patients boarding a lot longer in an emergency unit.—ID6

### Companions as consumers in the ED

Although physicians acknowledged various ways that companions as consumers contribute to and enhance practices in the ED, the shift to virtual companions also illuminated ways that physical companions interfered with different practices. One reason for why physicians expressed a preference for the virtual companionship introduced during COVID-19 was that they viewed patient treatment as their primary task and having virtual companions increased the efficiency of their clinical tasks. Specifically, due to fewer interruptions and less crowding while treating the patient, they felt they were able to prioritise patient care, resulting in fewer mistakes.

> Before there were two visiting times per day, an hour-long visiting time, which means that every patient in the unit could have two people within that hour, which literally triples the amount of people in the room. So, it's also just getting in the way, it's two hours of the day where you can't do anything because there's no point in being there.—ID5

> So not having those people to kind of judge you for what was going on definitely took off the stress of the work to some degree, because you could kind of get on with what you needed to do.—ID8

To compensate for the lack of physical presence, EDs implemented various practices such as contact sheets on arrival, tick sheets, involving other departments such as psychology or having a dedicated health provider conducting all the calls to families. Physicians described being able to communicate suggestions for different solutions to the consultant level, who worked with management to implement different systems. However, physicians felt these methods frequently resulted in calls not being conducted, or the physician completing the call themselves. It also created increased unrest among the virtual companions, resulting in a higher number of calls to the ED overwhelming their phone system, a phone system designated for emergent referrals. The in-person interruptions that occurred pre-COVID-19,

shifted to the phone, especially with many companions wanting to speak to the physician directly, and not other providers. The time-consuming nature of these changes resulted in the physician having less time for tasks related to patient care.

> The phone calls to the hospital were very difficult to handle because sometimes it'd be 22 family members of the same patient who all wanted a detailed update from the doctor. Each of which identified themselves as the family spokesperson.—ID3

Separate from the frequency of communication, physicians described the nature of communication with companions shifting throughout COVID-19 due to the severity and rapid changes in the patient's condition. Physicians acknowledged that dealing with companions virtually rendered these conversations more difficult. In addition to changing the nature of their communication with companions, it impacted how the provider interacted with the patient, including being more selective with their words and increasing compassion and empathy.

> It's probably one of the hardest things I've ever had to do to phone so many families and tell them that their family members died or was going to die or wasn't going to make it…I think I definitely learned to kind of find the compassion even when you're tired and exhausted.—ID8

### Finding balance for companion engagement

Beyond the time-consuming nature of updating virtual companions, physicians were able to identify both benefits and challenges associated with virtual companionship, throughout COVID-19 and within the context of the ED in general. They expressed that finding a hybrid of pre-COVID-19 practices and COVID-19 restrictions moving forward is necessary for providers, patients and companions. Physicians offered up different suggestions such as increasing the capacity to facilitate video calling, limiting the number of companions at the bedside, allowing one designated patient spokesperson at the bedside, reducing the length of visiting hours and having companions fill in up-to-date contact information when dropping off patients.

> We need to find something that works a bit better for families, but also allows us to actually see our patients timely and not have patients waiting for 12 hours to see a doctor, those kinds of things.—ID3

> We sometimes easily lack the human element in the emergency department when pressed for beds and patients and just chaos that is generally in the ED… There was across the board, from all my colleagues, a much more human aspect to that kind of [patient] interaction, than there were before.—1D7

Physicians mentioned that COVID-19 changes, specifically patients having no companions, illuminated inadequate practices in EDs that must balance humanity factors with the dynamic nature of emergency medicine. Physicians acknowledged finding a way for these factors to coexist is an important lesson to bring forward throughout and post-COVID-19.

## DISCUSSION

COVID-19 caused many formal and informal changes that providers had to juggle in conjunction with companion restrictions. Formal changes included structurally splitting the ED, different clinical protocols and prohibiting patient companions. Informal changes included disrupting tacit team knowledge, provider decision-making and communication adaptations. Physicians provided insight from their perspective on the impacts companion restrictions had on the ED, which can help inform future change and preparedness. Physicians were faced with a new scenario of no patient companions, yielding a paradox as the efficiency and quality of patient care simultaneously improved and suffered with virtual companions engaging as both providers and consumers in the ED. Emergency physicians' perceptions of the lacking physical presence of companions emerges as an understudied area, specifically in EDs, and should be incorporated into the greater debate on patient companions. The existing literature primarily focuses on companions from a patient-centred care perspective, as companions contribute to feelings of patient agency, safety and improved health outcomes.[11–13] Totten *et al*[13] found that 78% of patients in the ED felt their companion was important to their care for social support and advocacy.[13] Although there are no official guidelines for companions in the ED, there is a large body of literature in support of companion presence in other healthcare settings such as maternal care and the intensive care unit (ICU).[33 34] For example, the WHO has issued recommendations for companions of choice to be present during labour and childbirth and argues that having a companion they trust contributes to women feeling safe and strong, and leads to improved maternal and perinatal outcomes. One of the main factors they attribute this to is that companions bridge communication between the healthcare professionals and women.[34]

Overall, physicians acknowledged both benefits and challenges with virtual companions, and expressed the desire, in moving forward, to find a balance between the policies and practices pre-COVID-19 and during COVID-19. Iness *et al* and Herbst and Kuntz conducted studies that include the physician perspective in the ICU during COVID-19 and found that physicians perceived virtual companions and telecommunication to increase their workload and limit effective and empathetic communication.[20 35] This aligns with our findings that physicians noticed virtual companions to be more time-consuming. However, there are few studies that include the physician perspective, and none conducted specifically in the ED. Therefore, more research is needed to situate our specific findings, in order to propose comprehensive and sustainable solutions moving forward.

In addition to companion restrictions, physicians explained that COVID-19 patients were not initially moved to isolation facilities, but instead separated within the hospital into 'COVID-19 likely' and 'COVID-19 unlikely' areas in EDs based on their respiratory symptoms. The literature from other recent disease outbreaks such as SARS and Ebola, predominantly supports physical companion presence on the basis that it yields greater benefits to the patient than risks.[36] It is argued that restricting companions can discourage patients from seeking care, which can undermine infection control efforts, as found in the Ebola response.[37] When positioning our results within the literature on companion restrictions during disease outbreaks, our findings challenge whether hospitals and health systems have sufficiently applied lessons learnt about companions from previous outbreaks. There needs to be increased incorporation of companion restriction research for future emergency preparedness and response planning.

Participating physicians perceived their primary role to be healthcare providers, treating the patient and thought that having virtual companions was beneficial as it reduced interruptions, errors and stress. Results are quite divided from the limited ED literature on health providers' views of companions under normal circumstances. Timmermans and Meyers *et al*, propose that health professionals perceive companions as a hindrance to the process of saving a patient's life.[38] Generally, physicians were more worried than other health professions about the presence of companions in terms of crowding and interruptions. According to Axelsson *et al*, 61% of health professionals felt they would have increased anxiety and stress in life-saving procedures with companions' present.[38] The alignment of our findings with the literature illuminates the importance of including physician input when implementing policies, as they impact physician performance and subsequently patient care and safety. Further research needs to be conducted to find a balance which respects the physician's perspective, while also protecting patient-centred care.

Physicians perceived companions to be important for gaining collateral information about the patient. In the ED-focused literature, physicians primarily did not perceive companions to be associated with patient care, even though they too understand them to provide critical health information as the patient's spokesperson.[38] However, there is a knowledge gap regarding the patient care effects and potential added inefficiency associated with physical companions being switched to virtual communication. This gap could be attributed to the fact that most existing literature does not concern provider populations that have experienced companion restrictions before. Therefore, they have never actively had to consider how care and efficiency would shift when the companions switch to being virtual.

COVID-19 provided an opportunity for physicians to engage with reflexive monitoring as they appraise the role of companions, having had a prolonged experience without them. Moving forward, there is an opportunity for further research into whether these experiences have led to changes in how physicians perceive companions as an important component of care.

### Strengths and limitations

This study obtained in-depth knowledge of one professional base sampled at different time points from August 2020 to January 2022, which provides a sufficient period to capture changed practices.

This study faced challenges with recruitment attributed to the delays caused by the COVID-19 vaccine roll-out, which put research project approvals on hold. Due to the extenuating circumstances, this study had to pivot from purposive to convenience sampling, which can introduce a sampling bias caused by lack of equal access to participation, thereby affecting the transferability and trustworthiness of the results.[39] Additionally, these circumstances resulted in a smaller sample size, potentially introducing selection bias. However, to minimise the impacts, we chose to maximise saturation by focusing on the perceptions of one professional base. Our sample included experienced physicians, from multiple hospitals, who were able to compare practices before and during the pandemic. However, since COVID-19 is ongoing, we are not able to exhaustively capture how the physicians continue to perceive the changes. Additionally, we acknowledge the physician bias limits the transferability of this study.

This study combined two qualitative datasets that used different interview methodologies. This may have posed challenges for analysis because it introduced the potential for bias when coding dataset I. It did not involve the use of an interview guide, since the coder may then have been inclined to look for codes or themes that align with the main topics from the dataset II interview guide. This bias was reduced by continuously discussing the data among the research team. However, both studies focused on gathering perceptions from in-hospital emergency physicians. By combining these two datasets, with different types of data, we were able to capture a wider range of perceptions. In the voice recordings, physicians were able to share their perceptions in real time pre and post shift, allowing relevant situational information to be shared, that might have been forgotten by the time of a follow-up interview. We were able to compare both datasets, reinforcing our situational and reflexive data, which has strengthened the quality and dependability of our analysis and results in relation to our research aim.[31]

### Implications of the study

Learning from physicians' experiences and perspectives can help create a more holistic understanding of the contribution of the companion in the ED. This can inform companion policies that will promote preparedness, patient-centred care and improved health outcomes in the everyday context, in preparing for the continuation of the COVID-19 pandemic and in future disease outbreaks. It also provides a platform for future research to increase

emergency preparedness. Specifically, to capture more comprehensive results, the study can be continued with purposive sampling, increased sample size and the inclusion of more professional bases, such as nurses, to gain a wider perspective of the companion restrictions. In addition, follow-up studies can be conducted once COVID-19 has settled to learn how these changes were normalised.

## CONCLUSION

Physicians were introduced to the benefits and challenges associated with virtual companions. They felt the benefits of reduced crowding and interruption of bedside care and the challenges associated with the lack of collateral information and the time-consuming nature of virtual communication with companions. Overall, physicians expressed their hope that policies achieving a hybrid of pre-COVID-19 and COVID-19 practices will be sustained moving forward, including the use of contact information sheets, limitation of the number of bedside companions and increased access to video technology. By learning from their perceptions, there is an opportunity to better prepare the ED and providers for the next pandemic or other crisis that may affect companion policies, to maintain the highest level of patient-centred care.

**Acknowledgements** The authors wish to thank all the physicians who participated and shared their experiences. We would like to acknowledge Elzarie Theron and Helena Erasmus for their contribution to the data collection from dataset I.

**Contributors** All authors contributed to the conception and design of the study. LEW was the primary contributor to data collection, data analysis and interpretation, paper writing, revisions and submission. HMA contributed to the qualitative methodology design, data analysis and interpretation, as well as draft, and final submission revisions. WS contributed to sampling and recruitment, draft and final paper revisions, data sharing and providing the final approval for submission. All authors agree to be accountable for their work, with WS acting as guarantor.

**Funding** The authors have not declared a specific grant for this research from any funding agency in the public, commercial or not-for-profit sectors.

**Competing interests** None declared.

**Patient and public involvement** Patients and/or the public were not involved in the design, or conduct, or reporting, or dissemination plans of this research.

**Patient consent for publication** Not applicable.

**Ethics approval** This study involves human participants and was approved by the University of Cape Town's Faculty of Health Sciences Human Research Ethics Committee (HREC ref. 052/2021) (online supplemental appendix 2). The ethical application for dataset I (HREC ref. 386/2020) included approval for the anonymised data to be reanalysed for similar research questions (online supplemental appendix 3). All participants were provided with an information and consent letter, and they were able to ask any questions before signing (online supplemental appendices 4 and 5). Further approval was sought from the provincial department of health and the hospitals for participants to feel safe participating, with no negative repercussions in their workplace.

**Provenance and peer review** Not commissioned; externally peer reviewed.

**Data availability statement** Data are available upon reasonable request. Within the transcripts, physicians discuss personal and potentially identifying experiences from their workplace. The informed consent that all physicians signed promised full anonymity. Therefore, making the full data set publicly available would potentially breach their privacy. Following data requests, transcripts will be reviewed for any potential identifying information and will only be made available to researchers who sign a data sharing agreement. This might also be subject to further ethical review. Data requests may be sent to WS at willem.stassen@uct.ac.za.

**ORCID iDs**
Lauren E Wiebe http://orcid.org/0000-0003-3363-9735
Willem Stassen http://orcid.org/0000-0002-1486-4446

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
