## [Reviewer comments · BMJ Open]

ARTICLE DETAILS

TITLE (PROVISIONAL)	Companion Restrictions in the Emergency Department during COVID-19: Physician Perceptions from the Western Cape, South Africa
AUTHORS	Wiebe, Lauren; Alvesson, Helle; Stassen, Willem

VERSION 1 – REVIEW

REVIEWER	Debkumar Chowdhury Central Manchester and Manchester Children's University Hospitals NHS Trust
REVIEW RETURNED	27-Jan-2023

GENERAL COMMENTS	Reviewer comments for BMJ open-2022-070982  [ ] The authors have studied the impact of companion restrictions in the Emergency Department during COVID-19, a perspective from South Africa. [ ] During the peak of the COVID-19 pandemic, restrictions on families visiting patients were in place across the globe in an effort to reduce the transmission rate. The authors have thereby chosen an important topic that studies the psychological impact of these restrictions. [ ] The authors should be commended on approaching this topic from a South African perspective. Thank you for sharing the perspective with a global audience [ ] This was a qualitative study with a reflexive thematic analysis guided by the Normalisation Process Theory. [ ] The authors have correctly identified the limitation that the study was limited to one centre, and the external validity of the results could possibly be limited. [ ] The utilisation of two different datasets across two different time frames adds overall value to the article. [ ] The utilisation of a pilot study in Sweden prior to the study carried out in South Africa is duly noted to be positive. [ ] The authors' discussion and conclusion to the study are appropriate and adequately address the purpose of the study [ ] Could the authors kindly explain the statement on Page 7 Lines- 41-51- 'The literature' [ ] The sentence on Page 9 Lines 21-27 does not have a clear message, could the author kindly reframe the sentence to ensure that the message is lost [ ] Page 10 Line 7 could be re-written as 'The study was undertaken in the Western Cape, South Africa as this region historically had a higher proportion of presentations of emergent conditions than elsewhere in South Africa' kindly note this is only a suggestion [ ] Could the authors kindly ensure that restructuring of sentences is undertaken as many of the sentences within the article
---

	are rather long and wordy? I am certain that the authors would not want to risk losing the message that they want to convey to the reader. [ ] The sample size is limited, could the authors kindly explain what method was utilised to limit the impact of potential selection bias? [ ] Could the authors kindly ensure that grammatical errors are appropriately addressed especially the use of tenses? [ ] Whilst the inclusion of the responses is beneficial to the study, could this be placed in the appendix section? [ ] Could the authors explain in detail how to proceed with this study? What alterations would they consider in future studies?
--	--

REVIEWER	Yang Lyu Chinese Academy of Medical Sciences and Peking Union Medical College Institute of Basic Medical Sciences
REVIEW RETURNED	04-Feb-2023

GENERAL COMMENTS	Thank you for the opportunity to review this very interesting manuscript. The COVID-19 indeed has bring big challenges for companion restrictions in ED. The abstract of the result should be revised, the themes should be extracted, more concise and clear to let the readers directly know the findings from the interview. I recommend minor revision of the manuscript.
---

VERSION 1 – AUTHOR RESPONSE

Could the authors kindly explain the statement on Page 7 Lines- 41-51- ‘The literature’	Thank you for your comment. When referring to “the literature”, we were referencing the literature that was published on the topic of companions in the hospital, prior to the COVID-19 pandemic. We have adjusted the sentence in the manuscript to specify already published literature.
The sentence on Page 9 Lines 21-27 does not have a clear message, could the author kindly reframe the sentence to ensure that the message is lost	Thank you for highlighting this for us. We have rewritten those sentences to clarify our point.
Page 10 Line 7 could be re-written as ‘The study was undertaken in the Western Cape, South Africa as this region historically had a higher proportion of presentations of emergent conditions than elsewhere in South Africa’ kindly note this is only a suggestion	Thank you for your suggestion. We have discussed this with our colleagues in Cape Town and we have opted to keep our original statement.
Could the authors kindly ensure that restructuring of sentences is undertaken as many of the sentences within the article are rather long and wordy? I am certain that the authors would not want to risk losing. the message that they want to convey to the reader.	Thank you for the feedback. All three authors have reviewed the manuscript and edited sentences where appropriate to make them more succinct. Additionally, it has been edited by language editor.
The sample size is limited, could the authors kindly explain what method was utilised to limit the impact of potential selection bias?	We agree with your comment and have expanded on this in our limitations section.

Could the authors kindly ensure that grammatical errors are appropriately addressed especially the use of tenses?	We appreciate the feedback. All three authors have reviewed the manuscript to ensure grammar and tenses are consistent and appropriate. Additionally, it has been edited by language editor.
Whilst the inclusion of the responses is beneficial to the study, could this be placed in the appendix section?	Thank you for the suggestion. We have reviewed the quotes and made them more succinct where appropriate. We hope they serve a clear purpose now of understanding physicians' perceptions.
Could the authors explain in detail how to proceed with this study? What alterations would they consider in future studies?	Thank you. We have expanded on this within our implications section.

VERSION 2 – REVIEW

REVIEWER	Debkumar Chowdhury Central Manchester and Manchester Children's University Hospitals NHS Trust
REVIEW RETURNED	27-Mar-2023
GENERAL COMMENTS	Thank you to the authors for considering the changes as suggested in the initial review. I am thereby satisfied with this revised version as it addresses all the concerns that I had previously stated. Good Luck to the authors.